# Coevolutionary Analysis of Protein Subfamilies by Sequence Reweighting

**DOI:** 10.3390/e21111127

**Published:** 2019-11-16

**Authors:** Duccio Malinverni, Alessandro Barducci

**Affiliations:** 1Medical Research Council (MRC) Laboratory of Molecular Biology, Cambridge CB20QH, UK; 2Centre de Biochimie Structurale (CBS), INSERM, CNRS, Université de Montpellier, 34090 Montpellier, France

**Keywords:** coevolutionary analysis, direct-coupling analysis, specificity determining contacts, sequence reweighting, maximum entropy models, protein contact predictions

## Abstract

Extracting structural information from sequence co-variation has become a common computational biology practice in the recent years, mainly due to the availability of large sequence alignments of protein families. However, identifying features that are specific to sub-classes and not shared by all members of the family using sequence-based approaches has remained an elusive problem. We here present a coevolutionary-based method to differentially analyze subfamily specific structural features by a continuous sequence reweighting (SR) approach. We introduce the underlying principles and test its predictive capabilities on the Response Regulator family, whose subfamilies have been previously shown to display distinct, specific homo-dimerization patterns. Our results show that this reweighting scheme is effective in assigning structural features known a priori to subfamilies, even when sequence data is relatively scarce. Furthermore, sequence reweighting allows assessing if individual structural contacts pertain to specific subfamilies and it thus paves the way for the identification specificity-determining contacts from sequence variation data.

## 1. Introduction

The last decade has seen the emergence and maturation of coevolutionary methods aimed at predicting functionally interacting residue pairs from sequence alignments of homologous protein sequences [1,2,3,4,5]. The novel methodological developments, based on the use of global statistical models, have led to significant improvements in the quality of inter-residue contact prediction and computational structure prediction. This can be seen in the high scores obtained by the top-ranking teams in the recent CASP competition (Critical Assessment of Protein Structure Prediction), which mostly rely on coevolutionary predictions to guide their structural modelling [6]. Beyond the de-novo prediction of novel folds [5,7,8], coevolution-based analysis has also allowed the structural characterization of homo [2,9,10,11] and hetero-oligomeric [4,12,13] complexes and the determination of conformational ensembles [10,14,15].

The success of covariation-based contact prediction relies on the availability of deep multiple sequence alignments (MSAs) of homologous proteins. The rapid growth of protein sequence databases [16,17], driven by the decrease in cost of next-generation sequencing, resulted in the availability of very large protein families. In addition, the advent and improved access to meta-genomics databases further increased the pool of available sequence data [8]. In such a data-rich regime, the availability of ultra-large protein families (with typically more than 100 K homologous sequences) raises the intriguing question of how to analyze subfamily specific structural features by sequence covariation.

Indeed, coevolutionary analysis generally results in contacts predicted at the whole family level, thus predicting contact maps putatively formed by any member of the protein family. However, for large families consisting of multiple subgroups (or subfamilies), gene duplication and specialization lead to structural and functional variability of paralogous proteins sharing the overall same fold, but potentially carrying a sub-set of different contacts defining subfamily specificities [18,19]. Similarly, organisms with different genetic backgrounds and evolving in different environments will be subject to different fitness landscapes, thus not necessarily requiring the exact same structural and functional features, while still maintaining the same overall fold and function [20,21]. These observations imply that not all the members of a large protein family will necessarily satisfy all contacts predicted by coevolutionary analysis. Furthermore, the limited statistical weight of smaller subfamilies within a global alignment may prevent the identification of their specific features in a standard analysis.

This last point is of particular importance in the inspection or modelling of precise features pertaining to particular members of protein families rather than features common to the whole family. The latter scenario is typically encountered when dealing with complex eukaryotic families of great pharmacological interest, e.g., nuclear receptors [22] or G-protein coupled receptors [23], where the focus is often on specific members or sub-group. To generate predictions applicable in practice, one might need to understand or predict the effects of a small number of mutations affecting a particular set of sequences in the family. As such, novel tools and methodological developments designed for the analysis and identification of subfamily specific features at the contact level are currently required.

Some approaches have been already proposed to tackle this problem question. In [9], the authors propose to split the sequence dataset in multiple subfamily specific alignments and perform multiple independent Direct Coupling Analysis (DCA) on the sub-alignments, thereby successfully identifying subfamily specific features. Similarly, the authors in [24] performed independent DCAs on sub-class specific alignments to study different binding modes of protein complexes. Alternatively, a sequence reweighting scheme has been introduced in [10], whereby sequences belonging to two different phylogenetic groups where continuously reweighted and specific coevolutionary signals recorded, thereby assessing their phylogenetic origin. In a more recent development, authors of [25] introduced the use of restricted Boltzmann machines to simultaneously identify subfamilies and their characterizing motifs.

In the following, we build upon the reweighting concept introduced in [10], showing in a complex multi-dimensional case how this reweighting strategy can be used to identify sub-family specific contacts even in the case where the number of sequences is very low.

## 2. Results

To investigate how subfamily specific structural features can be extracted from complex protein families, we focus on the abundant and well-characterized family of bacterial response regulators (RR). RRs are part of the bacterial two-component signaling system, which forms one of the major transmembrane signaling systems in bacteria, and is generally composed of a transmembrane receptor and a cognate RR [26]. Upon sensing extracellular stimuli, the receptor usually auto-phosphorylates through its kinase domain and the phosphoryl group is then transferred to the RR. Prototypical RRs are generally comprised of a receiver domain, which are activated by the transfer of the phosphoryl group from the kinase domain of the receptor, and a C-terminal DNA binding domain, which acts as a transcription factor [27].

The interest in RRs as model system to study subfamily specific features originates in the fact that these proteins are classified according to their domain structure into different groups, which are characterized by alternate homo-dimerization interfaces [27]. As such, RRs form a (large) protein family, composed of several well-defined subfamilies displaying well-characterized different structural features

In particular, we focus here on three of the largest RRs subfamilies, namely the OmpR, LytTR and GerE classes, which share the same domain architecture (Figure 1A) but have been shown to exhibit different homo-dimeric arrangements. Interestingly, while RRs are classified according to the nature of the C-terminal DNA-binding domain, the receiver domain alone carries class-specific signatures, as clearly shown by a principal component analysis of its sequence space (Figure 1B). Indeed, we observe that sequences belonging to the three subfamilies form relatively well-defined clusters in the plane defined by the first two principal components, which bear the largest variance in sequence space. This finding suggests that a significant fraction of the sequence variability in the receiver domain can be explained by the nature of the tethered DNA-domain. The three subfamilies discussed here thus carry distinct sequence signatures, which pave the way for the investigation of class-specific structural features encoded in the sequence covariation of the receiver domain, as previously noted [9].

The high-resolution structure of proteins belonging to these three subfamilies have been determined, and in particular, models of their homo-dimeric arrangements are available. While the overall fold of the receiver domain is very similar in all the three classes, the homo-dimeric interfaces display striking variations (Figure 1C–F). Specifically, members of the LytTR and OmpR have different arrangements but their interfaces involve similar regions of the receiver domain, as shown by the corresponding contact maps (Figure 1C). In contrast, the homo-dimeric interface of the GerE subfamily involves a dramatically different region of the contact map, clearly highlighting a completely different binding mode. Furthermore, we note that only a small set of contacts can be actually used for defining the arrangement found in the LytTR class. Indeed, several residue pairs that are involved into this homodimeric interface form structural contact at the intra-molecular level in the whole family and hence cannot be used to define a subfamily specific feature.

For the sake of clarity, we will denote hereafter the structural interfaces corresponding to the OmpR, LytTR, and GerE as α-, β-, and γ-interface, respectively (Figure 1D–F).

The most straightforward way to look for sub-class specific contacts in a DCA framework is to split the global alignment in multiple sub-class specific alignments and perform independent contact predictions. The comparative analysis of the resulting DCA predictions can then indicate which contacts are exclusively predicted in particular sets of sequences, as already shown for the RR family [9]. While effective, this approach requires that all subfamilies are composed of enough sequences to yield sufficient statistical power to perform precise contact prediction by DCA. Even if this strict requirement is fulfilled for the RR family and its subfamilies (Figure 1A), it is certainly not the case in general. We thus first investigated how the number of available sequences affects the capability of correctly assigning subfamily specific features. To this aim, we randomly subsampled the three class-specific MSAs retaining a finite fraction *B_f_* of the original sequences and performed independent DCA predictions on these smaller alignments.

We first measured the overall prediction precision as a function of *B_f_*, by comparing the *N* highest-ranked DCA predictions with a common global contact map, comprising all the intra- and inter-molecular contacts observed in the three reference structures (Figure 2A). As expected from the large size of the RR family, even subfamily alignments yield excellent overall prediction quality, with precisions of 85–90%, if we make use of all available sequences (*B_f_* = 1). Remarkably, in this case the full RR alignment (union of the three subfamilies) does not yield any significant increase in precision, highlighting the probable saturation of the prediction quality. Nevertheless, decreasing the fraction of retained sequences rapidly leads to reduced precisions, and this effect is proportional to the total number of sequences in the alignment. Therefore, the gap between the results obtained for the family and those obtained for individual subfamilies initially increases for smaller *B_f_*, while eventually the precision collapses for all the alignments when the statistical power is too low (*B_f_* < 0.01). Note that at *B_f_* = 0.01, the full RR alignment is comprised of 861 effective sequences (see methods), which is still acceptable for performing high-quality predictions, as suggested by the overall precision using the family alignment (76%). At the same subsampling level, the DCA results for the OmpR subfamily are still partially reliable (precision ~64%) whereas the predictions obtained with GerE and LytTR sequences are of limited to no practical use (50% and 33%, respectively).

We then specifically evaluated the range of applicability of the alignment splitting strategy for extracting subfamily features by focusing on DCA predictions of the α, β and γ interfaces. The α-interface (defined by 16 homo-dimeric contacts) is generally well recovered even at low sequence fractions using the OmpR sequences (Figure 2B). Indeed, DCA of this subfamily alignments can identify up to 60% of the contacts defining the α-interface, and roughly half of this interface is recovered on average even at *B_f_* = 0.01. Reassuringly, α-interface contacts are never predicted using the GerE sequences for any subsampling even if the analysis of sufficiently large LytTR alignments yields some predictions in this interface. This result is not completely surprising, if we take into account the close proximity of the α- and the β-interfaces in the contact map. Nevertheless, the huge gap between the fractions of α-interface recovered by the two subfamily alignments makes unambiguous the assignment of the α-interface to the OmpR sequences.

Conversely, the assignment of the β-interface represents a much more difficult case (Figure 2C), due to its smaller area (11 contacts) and the limited size of the cognate LytTR subfamily. In practice, even using the full alignment, only a small fraction of the interface is predicted using the LytTR sequences, whereas no β-interface contacts are predicted using either the OmpR or GerE specific alignments. The case of the γ-interface is somewhat intermediate (Figure 2D). Indeed, while this is the largest interface (20 contacts), the cognate GerE subfamily consists of significantly less sequences compared to the large OmpR. Analysis of large sub-alignments (*B_f_* > 0.01) results into the prediction of ~15–20% of the γ-interface for the GerE to be compared with ~5% obtained in the case of either OmpR or LytTR sequences. This identification gap is further decreased by statistical noise as we decrease the fraction of analyzed sequences. At *B_f_* = 0.01, the interface assignment to a single subfamily becomes ambiguous. At such samplings, the overall precision (Figure 2A) lies between 30% and 60% depending on the family. Thus, the identification of subfamily specific contacts in these low sampling regimes would require dealing both with ambiguous interface assignments and with a potentially very large number of false-positive predictions even in the intra-molecular part, which lowers the overall confidence one can have in the interface predictions.

Taken together, these results illustrate that even if the splitting strategy works efficiently when sufficient sequence data is present in subfamilies [9], DCA predictions obtained with subfamily alignments might become unreliable and unable to identify subfamily specific structural features in the case of more common family sizes.

In order to circumvent this limitation, we present and discuss here an alternative scheme that does not imply the analysis of isolated subfamily alignments but instead relies on assigning arbitrary statistical weights to subfamilies within the full family alignment [10]. The core idea of this strategy is to monitor the dependence of inter-residue statistical couplings on the weights associated to subfamilies. Residue pairs whose coupling strength is strongly correlated with the weight associated to a particular subfamily will be identified as potential structural contacts specific to that subfamily. By keeping a mixture of sequences belonging to multiple sub-classes, the inference of model parameters directly controlling structural features shared by multiple subfamilies (typically intra-molecular contacts for the common fold) will benefit from increased quality of local statistics, therefore potentially helping to stabilize the overall prediction quality.

We propose the following algorithm, hereafter referred to as subfamily reweighting (SR) (see Methods for implementation details):
All sequences in a global alignment are subdivided into K subfamilies, indexed by k = {1,…,K}.All the sequences belonging to a single sub-class are assigned a common weight $\omega_{k}\;\in \left[0,1\right]$.DCAs are performed, assigning weights {$\omega_{k}$} to the sequences in the inference step, for a varying set of sequences weights.The relative change in coupling scores is measured on a set of contacts of interest, as {$\omega_{k}$} is varied.Subsets of residue pairs whose overall coupling strength is strongly correlated to the change in weights are identified as subfamily specific contacts.


Additionally, we can record the overall precision computed over the whole contact map for all values of the class-specific weights. This allows identifying the regions of weight space over which the precision, taken here as proxy of our confidence in the predictions, remains in a reasonable range.

We illustrate the use of the SR approach on the RR family discussed above, in the case where only 1% of sequences are sub-sampled (*B_f_* = 0.01). In this context, the SR procedure consists of the following steps: Sequences are grouped into three sub-classes corresponding to the OmpR, LytTR and GerE subfamilies. We assign all combination of weights in the range [0,1] in steps of 0.01 to the three subfamilies (see Methods) and perform DCA analysis for each set of weights. We then measure the overall precision and the coupling-scores for the α-, β-, and γ-interfaces, as a function of the subfamily weights and we report the results as triangle plots (Figure 3A–D). In this representation, the three vertices of the triangles correspond to the cases where we only keep sequences of one subfamily, while each interior point corresponds to a DCA performed with linearly interpolated weights.

We first focus on the overall precision computed over the complete contact map, which shows that the reweighting procedure results into a relatively high precision (typically above 70%) over a large range of relative weights (Figure 3A). Unsurprisingly, the quality of DCA results sharply decreases only in near vicinity of the vertexes and edges, which correspond to limiting cases where only one (vertices) or two (edges) subfamily are analyzed. In particular, the lowest precision is obtained when using only the smallest LytTR alignment (bottom-right vertex), consistently with what reported in Figure 2. Remarkably, we can explore regions of the weights-space relatively close to any extreme case while maintaining an overall precision of at least 70%, in strong contrast to the sharp precision drop obtained with subfamily alignments (Figure 2A and Figure 3A). This finding suggests that we can reliably interpret the DCA results obtained for weights in a large portion of the weights space.

We now inspect how the coupling-scores are affected by sequence reweighting, specifically focusing on the average coupling scores of the residue pairs defining the a priori known interfaces (Figure 3B–D) (see Methods). It appears strikingly that larger weights for the OmpR sequences correspond to higher coupling-scores over the cognate α-interface (Figure 3B). The trend is nearly linear with the orthogonal distance to the OmpR subfamily, indicating that the coevolutionary signal of the α-interface arises from the covariation encoded in OmpR sequences and it does not depend on the relative weighting of the two other subfamilies.

Interpreting the behavior of the average-coupling score over the β-interface is more difficult (Figure 3C). Indeed, there appear to be two local maxima, located both near the GerE and the cognate LytTR vertexes, with a non-monotonous behavior in the central part. This complex behavior does not lean itself to an easy interpretation and might be due to statistical noise. Indeed, the very low number of LytTR sequences in the sub-sampled alignment, combined with the relatively small number of contacts within the β-interface, may be responsible for this non-conclusive case.

In contrast, the average coupling-scores over the γ-interface display a nearly linear trend (albeit slightly tilted) with the orthogonal distance to the GerE subfamily (Figure 2D) and unambiguously identify this interface as a structural feature associated to the GerE sub-class. The clear result for the γ-interface obtained with SR approach is thus in sharp contrast with the more ambiguous assignation based on DCAs on the subfamily alignments (Figure 2D).

The success of the SR procedure in correctly characterizing both the α- and γ-interface as subfamily features, even with a limited amount of sequences, motivates us to test whether we can extend the same approach to assign individual contacts to specific subfamilies. This extension would greatly widen the range of applicability of the SR approach to protein families lacking any previous characterization of potential subfamily features.

To this aim, we can determine the coupling-scores of each residue pair as a function of the subfamily weights, analogously to what reported for whole interfaces in the triangle plots (Figure 3B–D). This information can then be used to devise scoring functions $F_{ij}^{k}$ that quantify how strongly a given contact (*I,j*), is associated to the subfamily *k* (see Methods). While many functional forms can be adopted to define the scores $F_{ij}^{k}$, here we limit ourselves to a proof of principle and we test a simple approach based on multilinear kernel functions (see Methods).

We then test if this strategy may be used for associating individual homo-dimeric contacts, taken from the union of the α-, β-, and γ-interfaces, to a specific subfamily.

To this aim, we sort all the contacts using the three subfamily specific scores and we inspect for each subfamily the highest-ranked ones, which are assumed to best represent subclass features (Figure 4A–C). If we limit ourselves to the ten top-ranked pair of residues, the predictions match the cognate structural interfaces reasonably well, even using our simple functional form for the scoring functions (8/10 for α/OmpR, 5/10 for β/LytTR and 8/10 γ/GerE cases, respectively). As in previous analyses, LytTR represents the most challenging case due to the smaller amount of available sequences and the smaller cognate interface (β-interface, 11 contacts), as is reflected in the lower fraction of correctly predicted interface contacts for this subfamily (5/10 vs. 8/10).

These promising results, although imperfect, highlight the ability of the SR procedure to identify single residue-pairs pertaining to specific subfamilies, thus being potential candidates of specificity determining contacts. While results are based on the use of the simplest possible single-contact scoring functions as proof of principle, the use of more sophisticated subfamily specific scores will potentially increase the prediction quality of the method.

## 3. Discussion

The sequence reweighting approach presented here, combined with coevolutionary contact prediction, allows the characterization and analysis of pairwise contacts which pertain to protein sequences belonging to specific subfamilies. Using the well characterized response regulator family as a prototypical proof-of-concept system, we showed that the SR approach is capable of correctly assigning subfamily specific interfaces, as well as identifying specificity determining contacts. In particular, the reweighted use of all classes allows for statistically robust results, even in cases where only limited sequence data is available.

In the present work, we relied on the a-priori knowledge of the subfamilies, based on their domain architectures. In principle, this supervised component of the algorithm could be replaced by a pre-processing step consisting in clustering the sequences and thereby automatically identifying subfamilies [28,29,30]. In combination with the SR procedure, this would allow a large-scale search over protein sequence databases (e.g., PFAM [17]) to identify families with significant structural diversity at the subfamily level. Such an automated procedure will inevitably introduce assignment noise whose consequences on the robustness of the identification of sub-family specific contacts will have to be systematically evaluate.

Furthermore, the identification of subfamilies and their associated specific contacts might be of valuable help in the context of homology modelling, specifically in the scenario where only structural models of remote homologs are present. In such cases, it is possible to erroneously impose some structural features of the template homolog, whereas the particular target belongs to a subfamily possessing some critical structural differences. As such, being able to identify the subfamily specific contacts by sequence analysis might allow to better guide the modelling step and/or improve the critical assessment of homology models based on remote homologs.

Additionally, many structural features of protein families are currently inferred by the analysis of available “representative” structures, often determined on model organisms. While such structural models are of great value, they might actually represent “snapshots” of the heterogeneous structural ensemble characterizing the complete protein family [20]. A computational tool aiming at highlighting potential deviations from the common structural scaffold defining the whole family could thus help identifying sub-classes with novel uncharacterized structural features and thus fruitfully complement structural biology approaches.

From a practical point of view, the SR algorithm is based on associating subfamily specific weights to the sequences in the inference step. We here relied on DCA, a popular method to predict structural contacts [6], but in principle, the SR procedure can be incorporated in any prediction approach based on optimizing a data-dependent objective function analogous to the pseudo-likelihood discussed here. In a wider context, SR can be seen as an instance of a transfer-learning approach [31], whereby we make use of the available sequences of the whole protein family to maximize the statistical power of the method, while adapting it to specific sub-problems. Such classes of algorithms, aimed at maximally exploiting the available data, are of great interest particularly when the available training data is limited, as in the case of eukaryotic protein families. In this scenario, analyzing particular subfamilies requires the efficient use all the data available for the whole family, even when focusing on questions pertaining to specific paralogous sub-groups.

While we here focused on the sequence reweighting for sub-family analysis, alternative reweighting schemes have been suggested to address the problem of phylogenetic and sampling bias in large MSAs [32]. The SR procedure could in principle be easily combined with these approaches to potentially improve the prediction quality.

The identification of specificity determining features in subfamilies of protein sequences is a longstanding challenge in bioinformatics [33,34,35]. In this context, SR can be extended to investigate specificity determining interactions, beyond the well-studied problem of identifying specificity determining positions (SDPs). Indeed, prediction of sub-class specific contacts potentially allows the prediction of more precise structural and functional features involving pairwise epistatic interactions, non-detectable by single-site SDP analysis.

## 4. Materials and Methods 

### 4.1. Sequence Data Collection and Pre-Processing

All sequence data was obtained from the PFAM database, release 32.0. We downloaded all aligned sequences for the Response regulator (RR) family (PFAM ID: PF00072), comprising 342,025 response regulator sequences of length *N* = 112 amino-acids. Additionally, the alignments for the DNA binding domains of three response regulator families where downloaded, namely the OmpR family (PFAM ID: PF00486), the LytTR family (PFAM ID: PF04397) and the GerE family (PFAM ID: PF00196).

Subfamily alignments of the response regulator domain where then built by selecting sequences from the response regulator alignment which possess either the OmpR (78,494 sequences), LytTR (14,883 sequences) or GerE (49,868 sequences) domains.

To reduce phylogenetic and sampling bias, and to simplify the reweighting procedure, the three alignments where filtered by identity, keeping only sequences in the alignments with a maximal pairwise hamming distance of 90%, using the hhfilter utility of the hhblits suite [36]. This resulted in 40,857 OmpR sequences, 12,082 LytTR and 33,344 GerE sequences. The sequences being pre-filtered by identity, the absolute number of sequences is therefore equivalent to the effective number of sequences (at a 90% identity threshold) in the current discussion.

To explore the effect of smaller datasets, sub-alignments were generated by randomly selecting a fraction B_f_ of sequences from the alignments. 200 random subsets where generated for analyzing the MSA depth effect (Figure 2). Three random subsets where generated for the reweighting analysis (Appendix A).

### 4.2. Structural Data Collection and Processing

The following structural models of the subfamily specific RR homo-dimers were collected (Table 1).

Inter-residue contacts were defined whenever pairs of residues had any heavy-atom distance below 5 Å. Given the close proximity of the α- and β-interfaces we chose here such a stringent contact definition threshold, lower than typically used in coevolutionary studies [2,9,37,38]. This ensures the definition of orthogonal contact interfaces. Indeed, increasing the contact threshold progressively widens the definition of interface contacts and ultimately results in partially overlapping interfaces (Appendix A).

The intra-molecular part of the contact map was defined as the union of the three intra-molecular maps from the three structural model. The three dimer interfaces were defined as all contacts not in the intra-molecular distance map for each model respectively.

In the three X-ray structures, the identification of biological assemblies is unambiguous and the corresponding homodimers do not display any noticeable break of symmetry. The three homodimers present in the asymmetric unit of 4cbv (β-interface) are characterized by the same set of intermolecular contacts according to our definition.

The contact maps were aligned to the RR multiple sequence alignment using the *mapPDB* tool from the dcaTools package [39] (available at https://gitlab.com/ducciomalinverni/dcaTools).

### 4.3. Direct-Coupling Analysis and Sequence Reweighting

Direct-Coupling Analysis (DCA) was performed using the asymmetric pseudo-likelihood inference method [37,40] as implemented in the lbsDCA package [39] using default inference parameters (available at https://gitlab.com/ducciomalinverni/lbsDCA). The method infers the parameters of the Hamiltonian of a generalized Potts model
$$P\left(s\right)=\;\frac{1}{Z}e^{\sum_{i=1}^{N}h_{i}\left(s_{i}\right)+\;\sum_{ij}^{N,N}J_{ij}\left(s_{i},s_{j}\right)\;}$$
where *s* = (*s*_1_,…, *s*_N_) denotes the amino-acid sequence of length *N*, *Z* denotes the normalizing partition function and h_i_ and J_ij_ are model parameters controlling the single- and two-site frequencies to be fitted to the data (see, e.g., [41] for a full review on DCA and its applications).

The inference is performed by numerically minimizing the regularized negative pseudo-log likelihood *l_PL_* of the data with respect to the model parameters {h_i_, J_ij_} (see, e.g., [37] for a detailed discussion of the pseudo-likelihood inference method).
$$l_{PL}=\;-\;\frac{1}{B_{eff}}\overset{B}{\underset{b=1}{\sum }}\omega_{b}\;log\left(\frac{\exp\;\left(\sum_{i}h_{i}\left(s_{i}^{b}\right)+\sum_{i<j}J_{ij}\left(s_{i}^{b},s_{j}^{b}\right)\right)\;}{\prod_{i}\sum_{a=1}^{21}\exp\left(h_{i}\left(a\right)+\;\sum_{j\neq \;i}\left(a,s_{j}^{b}\right)\right)}\right)$$
where *b* indexes the available sequences, $\omega_{b}$ denotes the weight associated to sequence *b* (see below), $B_{eff}=\sum_{b=1}^{B}\omega_{b}$ and *a* indexes the 21 amino-acids.

Here, we introduced subfamily specific relative weights such that $\omega_{b}$ = $\omega_{k}$ for all sequence *b* belonging to subfamily k∈{OmpR,LytTR,GerE}.

We further restricted the weights to sum to unity, i.e.,
$$\omega_{\mathrm{OmpR}}+\omega_{\mathrm{LytTR}}+\omega_{\mathrm{GerE}}\;=\;1$$

While this is not strictly necessary, the normalization allows for a straightforward mapping from the 3 dimensional weights space to a visualizable 2D space. The relative weights were then varied in steps of 0.01, including the border cases $\omega_{k}=\left\{0,1\right\}$.

The raw inter-residue coupling scores were computed by the Frobenius norm of the coupling parameters.
$$S_{ij}={\left\|J_{ij}\left(A,B\right)\right\|}_{A,B}$$
where the norm is taken over the 20 natural amino-acids, excluding the couplings involving the gap-parameter, following [42].

Finally, the coupling scores are given by the average-product corrected (APC) raw scores following [43], i.e.,
$${\tilde{S}}_{ij}=\;\frac{S_{ij}-\;S_{i\cdot }S_{\cdot j}}{S_{\cdot \cdot }}$$
where · denotes averaging over the relevant dimension.

All DCAs were performed using four threads per computation and took ~7 s each on a standard desktop workstation, resulting in a total computational time of roughly 10 h for generating the results presented in Figure 4.

### 4.4. Kernel Function Scoring

In the reweighting approach, each residue-pair is characterized by a series of coupling scores computed at different relative weights ${\tilde{S}}_{ij}\left(\omega_{\text{OmpR}},\omega_{\text{LytTR}},\omega_{\mathrm{GerE}}\right)$. In order to annotate each contact by a single scalar for each family of interest, we introduce the following multi-linear kernel functions.
$$\varphi^{k}\left(\omega_{\text{OmpR}},\omega_{\text{LytTR}},\omega_{\mathrm{GerE}}\right)=\{\begin{array}{l} \omega_{\text{OmpR}}\left(1-\omega_{\text{LytTR}}\right)\left(1-\omega_{\mathrm{GerE}}\right)\;\text{if }k=\text{OmpR}\;\\ \left(1-\omega_{\text{OmpR}}\right)\omega_{\text{LytTR}}\left(1-\omega_{GerE}\right)\;\text{if }k=\text{LytTR}\\ \left(1-\omega_{\text{OmpR}}\right)\left(1-\omega_{\mathrm{LytTR}}\right)\omega_{\text{GerE}}\;\text{if }k=\text{GerE } \end{array}$$

Such kernel functions have the desirable property and smoothly interpolate between these three border cases. As such, they allow to effectively compute a single weighted coupling score for each contact and each subfamily, which continuously assigns higher weights to coupling scores computed in realizations which weighted sequences of the subfamily more.
$$\varphi^{k}=\{\begin{array}{c} 1\;\text{if }\omega_{\text{k}}=1\;\\ 0\;\text{if }\omega_{i}=1\;\forall \;i\neq k \end{array}$$

In practice, to focus on the relative variation of coupling scores, irrespective of their absolute value, we subtract for each residue pair the average coupling score $<{\tilde{S}}_{ij}>$ (averaged over all weights triplets) before computing the kernel integral. This finally allows to define an average-corrected subfamily specific score for each residue-pair.
$$F_{ij}^{k}=\;\underset{\omega_{\text{OmpR}},\omega_{\text{LytTR}},\omega_{\text{GerE}}}{\sum }\begin{array}{c} \varphi_{}^{k}\left(\omega_{\text{OmpR}},\omega_{\text{LytTR}},\omega_{\mathrm{GerE}}\right)({\tilde{S}}_{ij}\left(\omega_{\text{OmpR}},\omega_{\text{LytTR}},\omega_{\mathrm{GerE}}\right)-\;<{\tilde{S}}_{ij}>)\;\\ \forall \;k\;\in \left\{\mathrm{OmpR},\;\mathrm{LytTR},\mathrm{GerE}\right\}\; \end{array}$$

## Figures and Tables

**Figure 1 entropy-21-01127-f001:**
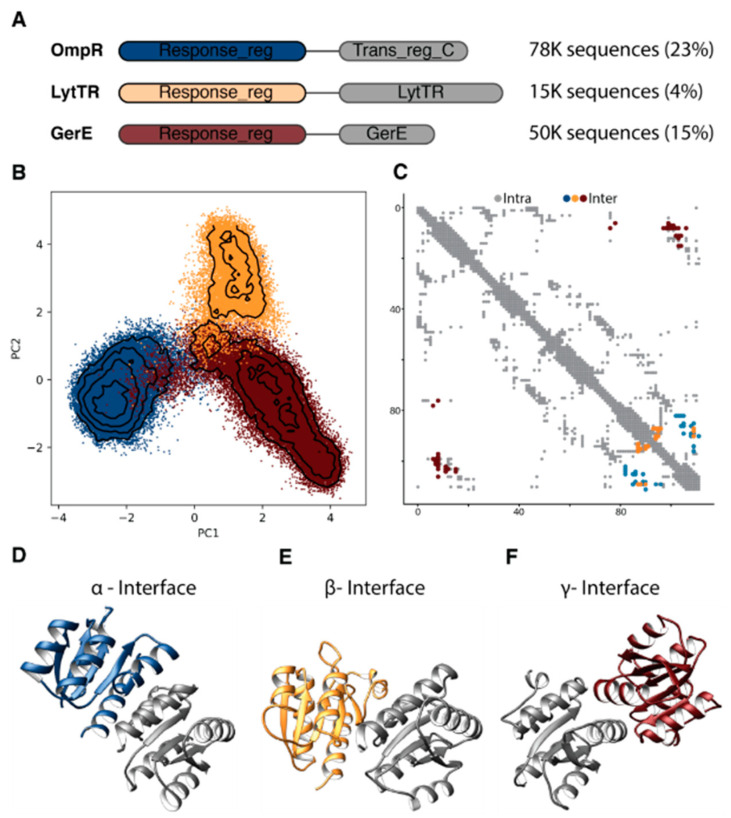
Sequence and Structural variability in the Response Regulator (RR) family. In all panels the color scheme follows the one defined in panel (**A**). (**A**) The three most abundant two-domain RR architectures with different dimerization modes, and their number of sequences in the complete RR alignment (fraction of the total number of sequences in parentheses). (**B**) Sequence variability of the RR family as shown by principal component projection of the RR sequences composed of the OmpR, LytTR, and GerE subfamilies. Projections are along the first two principal components. Black lines depict iso-density levels. (**C**) Contact map of three representative structures of the different subfamilies. Contacts are defined by a 5 Å distance-threshold between heavy atoms. Gray dots depict intra-molecular contacts. Colored dots depict homo-dimeric inter-molecular contacts (see Methods). (**D**–**F**) Heterogeneous homo-dimerization assemblies in the RR family. The three structural models used to define the contact map in panel C are depicted. The gray monomers in each model are structurally aligned.

**Figure 2 entropy-21-01127-f002:**
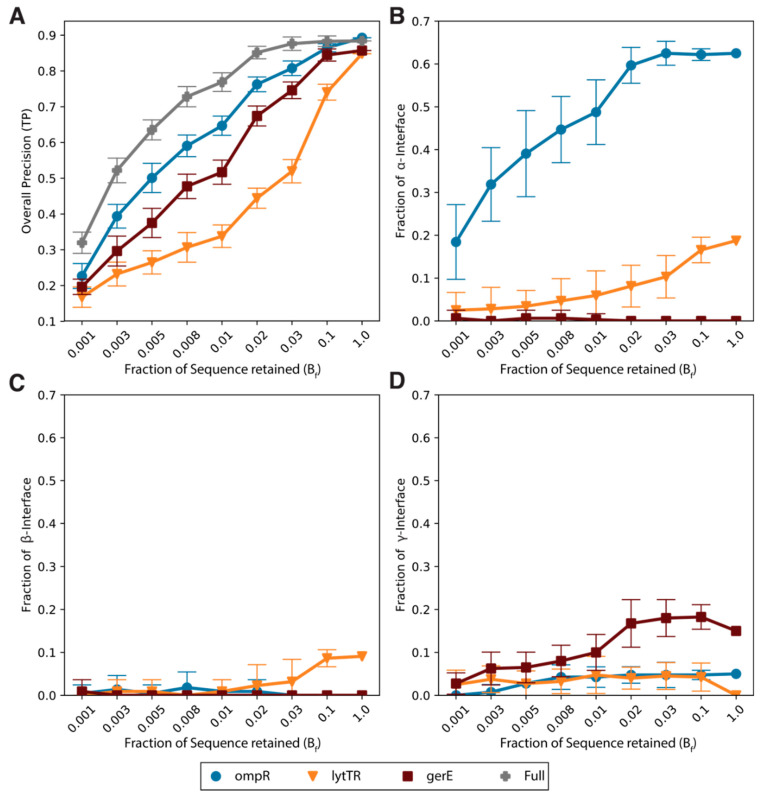
Prediction quality at varying alignments size. All reported quantities are shown as a function of the fraction of sequences randomly sampled from the full alignment B_f_. Error bars denote standard deviations over 200 random samplings. (**A**) Overall precision (i.e., true positive rate) computed over the complete contact map (union of intra-molecular contacts and all interface contacts). Full denotes the union of all three alignments. (**B**) Fraction of the α-interface predicted in the N (112) highest ranked contacts. (**C**) Fraction of the β-interface predicted in the N (112) highest ranked contacts. (**D**) Fraction of the γ-interface predicted in the N (112) highest ranked contacts.

**Figure 3 entropy-21-01127-f003:**
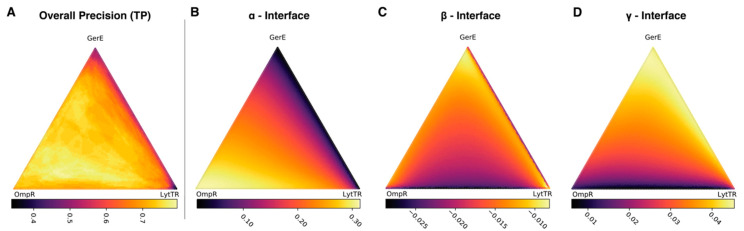
Results of sequence reweighting (SR). (**A**) Overall precision of the N highest ranked predictions, computed over the full contact map, comprising all intra- and inter-molecular contacts observed in the three reference structures. (**B**) Average coupling-score of the α-interface. (**C**) Average coupling-score of the β-interface. (**D**) Average coupling-score of the γ-interface.

**Figure 4 entropy-21-01127-f004:**
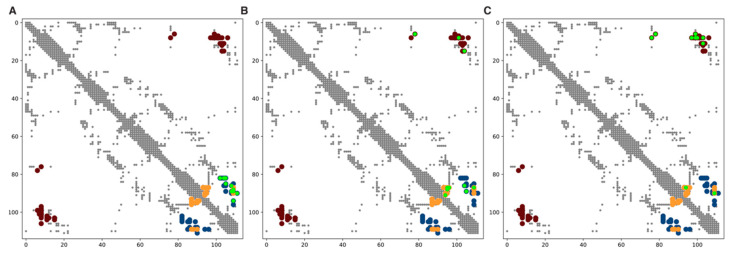
Identification of subfamily specific residue contacts by SR. Gray dots depict intra-molecular contacts. Colored dots depict interface contacts pertaining to the α- (blue), β- (orange) and γ- (brown-red) interfaces respectively. Dots in green are the top ranked contacts according to the $F_{ij}^{k}$ scores (see Methods). (**A**) Top 10 highest ranked SR contacts for k = OmpR. (**B**) Top 10 highest ranked SR contacts for k = LytTR. (**C**) Top 10 highest ranked SR contacts for k = GerE.

**Table 1 entropy-21-01127-t001:** Overview of used structural models.

Family	Interface	PDB ID	Model
OmpR	α-interface	1nxs	Biological Assembly 1
LytTR	β-interface	4cbv	Biological Assembly 1
GerE	γ-interface	4e7p	Biological Assembly 2

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
