# Peer review of "Coevolutionary Analysis of Protein Subfamilies by Sequence Reweighting"

_entropy, 2019, doi:10.3390/e21111127_

Round 1

Reviewer 1 Report

The authors, Duccio Malinverni and Alessandro Barducci, of the manuscript “Coevolutionary analysis of protein subfamilies by sequence reweighting”, take aim at the question if contacts predicted from coevolutionary analysis of a large protein families pertain to specific members of such pool. In this manuscript, the authors suggest a significant improvement of the contact prediction by a new approach using weighted alignment of analyzed sequences. Unlike the conventional prediction methods, in this scheme a global alignment of the whole family is performed that is re-weighted on the sub-family basis.

Although the aim of this study is well defined and the experiments are well planned, the manuscript suffers from two issues:

It is not stated if the dataset considered in this study included structure models with non-crystallographic symmetry (if the structures were solved using NCS operators). Many enzymes, although build as complexes of symmetric subunits, display breaks of symmetry that might result in altered interactions between the subunits. This issue should be critically addressed.

In conclusion, the authors suggest a timely study with important observations. The manuscript is very clearly and understandable. The authors went long way to make their scientific argument to an important problem in predicting complex protein-assemblies. Very positive, the authors clearly describe the limitation of the method. Therefore, suggest the manuscript to be accepted after minor revision.

Reviewer 2 Report

Isolating and accounting for sub-family specific constraints is extremely important for most applications of residue-residue coupling analyses but is a technically challenging problem given the large numbers of sequences required for these analyses. In this very well written and thoughtful study, the authors build on some of their previous results to look at how sub-family specific weights can be used to identify sub-family specific contacts. I am very positive on the manuscript, found it logical and easy to follow, and think that a fairly broad audience would understand the problem and solution (even if some details are slightly technical / specialized for the audience of researchers operating in this space). I have only very minor comments to contribute.

The PCA plot in Fig. 1B makes it clear that any decent clustering algorithm should do a good job at selecting three clusters and assigning sequences accordingly. However, there will be some noise in this process. An area for future research may be to show how/whether results vary when using a de novo clustering algorithm to select the number of sub-families and (noisily) assign sequences to them. I suspect that it wouldn’t affect things much here, but this also might be a particular clear case of sub-families compared to others. I don’t think this requires any re-analysis, but this robustness question might be worth noting a bit more thoroughly in the discussion and could/should be an area for future research. I personally don’t think it’s a huge deal, but it’s probably worth addressing run-time / computational resource constraints briefly in the discussion. Varying the weight for each sub-family in increments of 0.01 from 0 to 1 of course entails running a lot of DCA model inferences. These models have gotten better/faster, but is run-time a concern? How long would the analysis of Fig. 4 be reasonably expected to take? My most important comment is that sub-family specific weights will over-ride more commonly used phylogenetic weighting schemes used in DCA and associated methods. In our most recent work (https://www.mdpi.com/1099-4300/21/10/1000) we showed that commonly used sequence re-weighting approaches do not dramatically improve the predictive accuracy of DCA, but they nevertheless help (with the commonly accepted 80% sequence identity-based re-weighting being basically one of/the best). Honestly, I think this current paper is quite good and the authors have done enough to make a valuable contribution but I think mentioning that combining sub-family specific weights with more commonly used phylogenetic weights could furhter improve results. Conceptually, it doesn’t seem that it would be too difficult to try in the future. The authors note in the methods that they reduce phylogenetic bias by throwing away sequences with >90% identity but this is a pretty crude approach and their results might be improved by combining phylogenetic and sub-family weights. Additionally, most folks in the field will be commonly acquainted with phylogenetic weights so discussing how sub-family weights could be combined with these weights would be helpful. Journal information for reference 24 is missing/incomplete Font formatting on Page 2, line 75 gets weird (I think?). It looks slightly off, maybe a few sizes too small or something? Page 4, line 125 “three structures representatives” => “three representative structures” The nomenclature “Bf seems a bit unnecessary at least in the figures. It’s a personal preference but I think the figure would be easier to immediately grasp by a reader if the x-axes were simply labeled “Fraction of sequences retained (Bf)” Page 6 line 186, “…shown as function…” => “…shown as a function...” Page 2 line 67, define “DCA” at its first use as “direct coupling analysis (DCA)” Page 6 line 195, “sufficiently sequence” => “sufficient sequence” Page 9 line 279, “a-, b-, and g-” should be “alpha-, beta- and gamma-”? Page 9 line 277, missing a period at end of sentence. One reference in this general space that I found to be missing from the manuscript was: https://www.pnas.org/content/114/34/9122.short These authors look specifically at structural variability as one of the main determinants that apparently limit the overall accuracy of DCA methods (finding that many “incorrect” DCA predictions of residue-residue contacts are actually “true” contacts in different representative structures). Might be worth referencing but of course there are a million papers in this space and to each their own! I don’t actually think that a 5 angstrom heavy atom cutoff is that strict. In our work (https://peerj.com/articles/7280/) we found that a stricter 4.5 angstrom cutoff for heavy atom distances corresponds roughly to a 7.5 angstrom beta-carbon based cutoff. We also found that only using the side-chain heavy atoms resulted in better predictions overall (when ignoring residue-residue pairs that are close in linear space on the amino acid chain). Not sure that anything needs to be added/changed, just making the observation for the authors if they’re unaware of this work since we found that the choice of method and cutoff can actually affect these DCA studies and is often overlooked.
